# MASLD: Prevalence, Mechanisms, and Sex-Based Therapies in Postmenopausal Women

**DOI:** 10.3390/biomedicines13040855

**Published:** 2025-04-02

**Authors:** Ilaria Milani, Marianna Chinucci, Frida Leonetti, Danila Capoccia

**Affiliations:** Department of Medico-Surgical Sciences and Biotechnologies, Faculty of Pharmacy and Medicine, University of Rome La Sapienza, 04100 Latina, Italy; chinuccimarianna@gmail.com (M.C.); frida.leonetti@uniroma1.it (F.L.); danila.capoccia@uniroma1.it (D.C.)

**Keywords:** steatosis, metabolic dysfunction, menopause, sex hormones, GLP1-RAs, estrogens

## Abstract

Metabolic dysfunction-associated steatotic liver disease (MASLD) is the most common chronic liver disease influenced by genetic, lifestyle, and environmental factors. While MASLD is more prevalent in men, women are at increased risk after menopause, highlighting the critical pathogenetic role of sex hormones. The complex interplay between estrogen deficiency, visceral fat accumulation, metabolic syndrome (MetS), and inflammation accelerates disease progression, increases cardiovascular (CV) risk, and triggers a cycle of worsening adiposity, metabolic dysfunction, and psychological problems, including eating disorders. Weight loss in postmenopausal women can significantly improve both metabolic and psychological outcomes, helping to prevent MASLD and related conditions. This review examines the prevalence of MASLD, its comorbidities (type 2 diabetes T2D, CV, mental disorders), pathogenetic mechanisms, and pharmacological treatment with GLP-1 receptor agonists (GLP1-RAs), with a focus on postmenopausal women. Given the use of GLP1-RAs in the treatment of obesity and T2D in MASLD patients, and the increase in MetS and MASLD after menopause, this review analyzes the potential of a stable GLP-1–estrogen conjugate as a therapeutic approach in this subgroup. By combining the synergistic effects of both hormones, this dual agonist has been shown to increase food intake and food reward suppression, resulting in greater weight loss and improved insulin sensitivity, glucose, and lipid metabolism. Therefore, we hypothesize that this pharmacotherapy may provide more targeted therapeutic benefits than either hormone alone by protecting the liver, β-cells, and overall metabolic health. As these effects are only supported by preclinical data, this review highlights the critical need for future research to evaluate and confirm the mechanisms and efficacy in clinical settings, particularly in postmenopausal women.

## 1. Introduction

Metabolic dysfunction-associated steatotic liver disease (MASLD), previously known as non-alcoholic fatty liver disease (NAFLD), is the most common chronic liver disease and is now included in the 2023 consensus definition of steatotic liver disease (SLD) [1,2]. MASLD is characterized by abnormal fat accumulation in the liver, accompanied by at least one of five cardiometabolic risk factors. This condition can progress from simple steatosis to more severe stages, such as metabolic steatohepatitis (MASH), cirrhosis, and hepatocellular carcinoma (HCC) [2,3]. Its global prevalence has increased by nearly 50% over the past three decades, reaching 38.2% [3], making it the fastest-growing cause of liver-related diseases [3,4]. The increasing prevalence of MASLD is largely driven by the parallel epidemics of related metabolic conditions such as type 2 diabetes (T2D) and obesity. MASLD also increases the risk of extrahepatic complications, such as cardiovascular events, chronic kidney disease (CKD), and extrahepatic tumors [2,3], contributing to its growing global health burden [2].

The development of MASLD is influenced by multiple factors: insulin resistance and conditions related to metabolic syndrome (MetS), such as obesity and T2D, are pivotal to its onset [5]. However, genetic predisposition, lifestyle habits [6], chromosomal factors, and especially age-related changes in sex hormone levels, also play an important role in metabolic disorders [7,8]. These factors may help explain why, despite a higher incidence of MASLD in men, conditions such as polycystic ovary syndrome (PCOS) and menopause increase susceptibility to MASLD/MASH in women [9,10,11,12]. Due to the significant cardiometabolic impact of MASLD, projections suggest that its clinical burden will increase substantially over the next 30 years without effective treatments [13]. Specifically, while MASLD has traditionally been more common in men, estimates predict that by 2030 both MASLD and MASH will increase in both sexes, with prevalence rates becoming more comparable, particularly for MASH (18.90% vs. 18.41%) [14]. The rise in visceral adipose tissue (VAT) and the adverse changes in carbohydrate and lipid metabolism during the peri-menopausal period, resulting from the cessation of ovarian function [5,15], can contribute to higher levels of inflammatory biomarkers, worsening disease progression, and impacting quality of life [16]. Obesity, particularly central obesity, is a key driver of MASLD, with insulin resistance, adipocyte dysfunction, and inflammatory markers playing a critical role in the development of MASLD and related conditions [17,18].

In morbid obesity, indexes of insulin resistance have been recognized as markers of histological severity of liver disease [19]. This may explain why the incidence and severity of diseases such as liver fibrosis increase after menopause [5], with women affected by MASLD experiencing more years of disability due to the disease than men [16].

Early detection, risk assessment, and management are essential to effectively address MASLD [13]. Furthermore, understanding the interaction between sex/gender and traditional risk factors can improve patient outcomes [20,21], especially within the evolving field of precision medicine, which is increasingly focused on sex/gender differences. The aim of this review is to explore the influence of sex, hormonal status, and the physiological and psychological changes associated with menopause on the prevalence, pathogenetic mechanisms, and treatment of MASLD, with a particular focus on GLP-1 receptor agonists (GLP1-RAs) pharmacotherapy. The main objective is to evaluate the potential of GLP-1–estrogen conjugates as an innovative therapeutic approach for this subgroup. In particular, emerging treatment strategies that combine the protective effects of estrogen with GLP1-RAs show promise in reducing weight and addressing metabolic dysfunction associated with menopause, providing targeted treatment options for MASLD and its complications in postmenopausal women.

## 2. Sex Differences in the Prevalence of MASLD and Related-Liver Complications

MASLD and MASH exhibit sex differences that vary with age, affecting both their prevalence and progression [22]. During the reproductive years, men typically have a higher prevalence of MASLD than women [12,23,24,25], progressing to more severe stages, such as MASH, cirrhosis, and HCC [7,8,12,26]. Under age 50, factors such as triglycerides (TG), LDL cholesterol, and platelet count are associated with liver health in men but not in women, and non-invasive fibrosis scores tend to be more reliable for assessing liver status in men [27]. Men also show higher levels of liver steatosis compared to premenopausal women [28,29]. However, after age 50, the prevalence of MASLD in women becomes similar to that in men [5,6,26,30,31]. This shift has led the latest joint European Association for the Study of the Liver (EASL)—European Association for the Study of Diabetes (EASD)—European Association for the Study of Obesity (EASO) guidelines on MASLD to emphasize that men over 50, postmenopausal women, and individuals with multiple cardiometabolic risk factors are at higher risk for progressive fibrosis, cirrhosis, and related complications [2]. By 2040, MASLD is expected to affect more than half of the adult population, with the greatest increase in women [32], who are 2.4 times more likely to develop MASLD after menopause [2,33], particularly between the ages of 50 and 60 years [34]. Furthermore, postmenopausal women with obesity and steatosis tend to have worse clinical outcomes [35], especially with more severe MASH fibrosis compared to premenopausal women [36], a key prognostic factor in severe MASLD that is difficult to reverse [5]. It has been reported that while women in general have a 19% lower risk of liver steatosis, they have a 37% higher risk of advanced fibrosis, especially after the age of 50 [37]. In addition, among people with T2D, postmenopausal women show a twofold increase in significant fibrosis compared to premenopausal women, while fibrosis rates in men remain similar regardless of age [38].

Women also experience a higher mortality rate from cirrhosis than men, suggesting that they may be more susceptible to severe forms of the disease [36]. Animal models of induced fibrosis show that while males are more prone to severe fibrosis in the early stages, prolonged liver injury in females leads to more active hepatic stellate cells (HSCs), accelerating fibrosis progression [39]. This difference in progression with age may explain the higher cardiovascular (CV) risk [7,40], and increased mortality rate in women compared to men [36,41]. Notably, MASH-related liver disease has become the leading cause of liver transplantation in women without HCC [42], and the second most common cause in men [9]. A large cohort study of 761,403 patients with MASLD confirmed that women have a higher incidence of all adverse liver events and a significantly shorter time to develop cirrhosis, making MASLD the leading cause of liver transplantation in women. Men, on the other hand, have a higher risk of CV disease, HCC, and non-liver cancers. However, this analysis was limited to patients with private health insurance in the United states [43].

In liver cancer, MASH has surpassed alcohol-related liver disease as the leading cause of HCC [42], with men generally experiencing rates two to four times higher than women [44,45]. However, a large meta-analysis found no significant differences in HCC incidence between men and women [46], although women are more likely to develop non-cirrhotic HCC [12,47]. The global increase in HCC associated with MASLD is driven by the increase in metabolic diseases [48], with confirmed risk factors including T2D, obesity, and the patatin-like phospholipase domain-containing 3 (PNPLA3) gene variant [49]. The interaction between the female sex and the PNPLA3 p.I148M variant [50,51], and the increasing prevalence of T2D and central obesity in postmenopausal women [31], may explain why the incidence and mortality rates of HCC in women are rapidly approaching those of men [52,53]. For example, although an Italian study observed a significant increase in MASLD-related HCC cases from 3.6% in 2002 to 28.9% in 2019, with a higher incidence in men, it identified overweight/obesity, hyperglycemia, and T2D as key risk factors for HCC [54]. In addition, a recent meta-analysis of 15,377 patients with MASLD-related HCC, found no significant sex differences [55], while US cancer statistics showed an increase in MASLD-related HCC cases in both sexes aged 50 and older, with a decrease in younger men [52]. Similarly, a Swiss study found that the percentage of MASLD-related HCC cases in women increased significantly from 21% (1990–1994) to 68% (2010–2014), with no such increase observed in men [56]. These findings suggest a shift toward a reduction in the male predominance of MASLD-related HCC [52,53,56].

A meta-analysis of 557,614 patients found that women with MASLD have higher rates of all-cause mortality, CV mortality, and cancer than those without MASLD, and a 44% higher risk of cancer than men with MASLD [57]. Older women are more likely to experience cerebrovascular events, which helps explain the increase in the number of female liver donors over the past 20 years, while the number of male donors has decreased [58]. The greater health vulnerability of older women, coupled with the higher prevalence of MASH, have been identified as factors that reduce the likelihood that women will receive a liver transplant due to increased surgical risks and a higher chance of dying while on the waiting list [59,60]. In addition, current stratification scores, such as the Model for End-Stage Liver Disease score, tend to underestimate mortality in women [60,61], leading to the proposal of sex-adjusted scores to address these gender differences in access to liver transplantation [61].

For extrahepatic cancers, associations have been observed between Fatty Liver Index (FLI) scores (30–60 and ≥60) and the presence of colorectal adenomas or early colon cancer in men [3,62,63,64], and breast cancer in postmenopausal women [3,63,64,65]. However, higher fibrosis scores have been linked to the development of all cancers [64], and especially in women, MASLD has been found to be a risk factor for colorectal adenomas, particularly when associated with liver fibrosis [63,66]. MASLD is also an independent risk factor for advanced metachronous colorectal neoplasia in women, but not in men [63,67], suggesting a stronger effect on colorectal carcinogenesis in women [67].

Given the critical role of sex and age in the development of MASLD and related cancers, identifying those at the highest risk is essential for effective screening programs and interventions [49]. These sex differences in the epidemiology of MASLD warrant further study, particularly focusing on postmenopausal changes, as these have a significant impact on disease progression and risk in women.

## 3. Gender Differences in Extrahepatic Diseases

MASLD is recognized as the liver manifestation of MetS, which includes factors such as visceral obesity, hypertension, insulin resistance, and dyslipidemia [68]. However, the relationship between MASLD and MetS is bidirectional, as MASLD itself contributes to the development of obesity, T2D, and dyslipidemia by affecting carbohydrate and lipid metabolism [5,15]. This explains the strong association between MASLD and extrahepatic diseases such as T2D, obesity, and CV risk [2]. Although obesity is more common in women, they appear to have some protection against obesity-related cardiometabolic conditions, including MASLD, up until menopause [41]. However, after menopause, visceral obesity, T2D, dyslipidemia, and MASLD contribute to increased CV risk in women [5]. There is also increasing evidence of a link between MetS, MASLD, and mental health problems, with possible gender differences [69]. Understanding the differences in MASLD-related comorbidities between women and men is essential for effective prevention, early diagnosis, treatment, and addressing the healthcare burden of liver disease [70].

### 3.1. Type 2 Diabetes and Obesity

Obesity, especially visceral obesity, and T2D are significant risk factors for MASLD and contribute to its progression to advanced fibrosis, cirrhosis, and HCC [2]. MASLD is more common in people who are overweight or obese, with a global prevalence of about 50% [71]. The prevalence of obesity varies by region [3,4] and is higher in women than men [72,73,74] across all age groups, with the highest rates observed between ages 50 and 65 [73].

A meta-analysis [75] found that 80% of people with MetS and obesity had MASLD, compared with only 18% of those without these conditions. Specifically, there is a strong positive correlation between MASLD and body mass index (BMI), emphasizing that both overweight and obesity increase the risk of MASLD in both men and women [34,76].

However, although a BMI ≥ 31.9 kg/m^2^ is associated with a significantly higher risk of MASLD in men compared to women, the prevalence of MASLD increases dramatically in postmenopausal women [34]. This increase may be due to an increase in visceral obesity in postmenopausal women. A cross-sectional study in Ghana found that the prevalence of MASLD and MetS in postmenopausal women was 49.48%, compared with 29.55% in premenopausal women. Notably, abdominal obesity was identified as the most common component of MetS in this population, significantly increasing the risk of MASLD [77]. A meta-analysis of over 24 million people found that the association between obesity and MASLD was less pronounced when BMI > 30 kg/m^2^ was used as a measure of obesity. This suggests that other measures of central obesity may be more effective in predicting liver outcomes, particularly in women [78]. Indeed, an increase in VAT was observed with age in Chinese adults, and BMI was not able to predict the proportion of VAT. Furthermore, in women, all measures of adiposity (VAT, subcutaneous adipose tissue SAT, BMI, and waist circumference WC) increased with age [79]. This underscores the importance of considering the diverse distribution of abdominal adiposity indices when assessing cardiometabolic risk associated with metabolic disease.

For example, in premenopausal women, no significant differences were found between adiposity indices such as BMI, WC, and waist-to-hip ratio (WHR). However, in postmenopausal women, the combination of these indices significantly improved the ability to predict MASLD, with WHR emerging as the most useful discriminator [80]. The Visceral Adiposity Index (VAI), which includes factors such as WC, BMI, TG, and high HDL cholesterol, was also identified as a marker for T2D, MetS, and CV risk. The hazard ratio for MASLD incidence in the highest VAI quartile was found to be 3.69 in men and 4.93 in women [81]. In addition, a positive correlation has been found between the FLI and MASLD, especially in women aged 40–64 years [82]. Since VAI serves as a marker of visceral fat dysfunction and distribution, one of the proposed hypotheses, the “portal theory” suggests that visceral fat has direct toxic properties on the liver [82]. These findings support that while men are susceptible to visceral fat deposition throughout life, women become more susceptible after menopause, likely due to the loss of the protective role of SAT and estrogen [12,22,83,84].

While chronological aging and obesity contribute to the development of T2D, ovarian aging and menopause play a significant role in the onset of glycemic dysregulation, further exacerbating the risk [85]. T2D and MASLD often coexist, with MASLD affecting up to 70% of people with T2D. These two conditions have a bidirectional relationship, with each influencing the progression of the other [86]. A study of middle-aged and elderly Chinese individuals found a stronger association between MASLD and T2D in men, likely due to higher levels of adiposity (BMI, WC, WHR) in men compared to women [87]. Accumulation of fat in the liver impairs glycogen synthesis, increases gluconeogenesis [88], exacerbates insulin resistance, and promotes the systemic release of proinflammatory cytokines and hepatokines [89]. While women of reproductive age tend to be more insulin sensitive, less prone to store fat in visceral and ectopic compartments, and have a greater capacity to secrete insulin and incretins [30,90], postmenopausal women are at higher risk for central obesity, hypertriglyceridemia, and prediabetes. These factors, which are characteristic of MetS, T2D, and the diagnosis of MASLD, increase the risk of both conditions in postmenopausal women [91,92]. A meta-analysis of middle-aged individuals found that T2D was significantly associated with the incidence of severe liver disease, with a more than twofold increase in the outcome of severe liver disease [78].

MASLD is a stronger risk factor for developing T2D in premenopausal women than in postmenopausal women or men, with the effect varying by severity [93,94]. The addition of MASLD to traditional risk factors improves the ability to predict the risk of incident T2D in both sexes, with the highest discriminative power seen in premenopausal women [93]. This suggests that MASLD contributes to the loss of biological protection against T2D in premenopausal women [95].

Conversely, women with dysglycemia have a similar [96], or even higher risk of developing MASLD compared to men. Specifically, an association between dysglycemia, female sex, and age ≥50 years was found in patients with MASLD, suggesting that active screening for hepatic steatosis would be recommended in younger women with any degree of dysglycemia [96]. A Chinese study of people with T2D (mean age 61 years) found a 59.36% prevalence of MASLD, with 1.43% of participants having cirrhosis. In particular, women with T2D had a higher risk of MASLD compared to men [76]. Research also suggests that prediabetes and T2D significantly increase all-cause mortality, but this effect was only observed in women [97]. The worsening of anthropometric and metabolic risk factors (such as homeostasis model assessment of insulin resistance (HOMA-IR), C-reactive protein (CRP), and lipid profile) may help explain how impaired glucose homeostasis contributes to the development of MASLD [98].

A recent global epidemiologic analysis showed that women with MASLD have a higher rate of T2D than men. However, the incidence of both conditions is influenced by the reproductive stage of life, with higher rates observed in prepubertal men and postmenopausal women [99]. Another study involving over 200,000 MASLD patients (52.8% women) found that women diagnosed with MASLD at an older age had higher rates of comorbidities such as MetS, hypertension, T2D, and obesity compared to men. In addition, women had higher rates of cirrhosis in both sexes [70]. This underscores the importance for healthcare providers to consider the close relationship between these conditions [86], and the influence of sex and age on the development, progression, and prognosis of MASLD.

Obesity is a well-established risk factor for T2D and MASLD, but some people, often referred to as having “lean disease”, develop these conditions despite having a BMI < 25 kg/m^2^. In these cases, factors such as abnormal fat distribution (especially increased visceral fat), sarcopenia, and early-stage β-cell insufficiency are significant contributors [86]. Sex analysis revealed that women with lean MASLD have a higher prevalence of MetS than men. They also have higher levels of TG, alanine aminotransferase (ALT), and gamma-glutamyl transpeptidase (GGT) than their overweight/obese counterparts. Given the higher average number of women in the lean MASLD group, the study suggests that menopausal or postmenopausal status may play a significant role in the development of MetS-related factors, making lean MASLD a more harmful phenotype [100] in postmenopausal women [101]. High testosterone levels in men may help explain their lower risk of developing T2D and insulin resistance [102]. Interestingly, another study found a higher incidence of lean MASLD in men than in women. However, both sexes showed an increase in incidence with age, with the peak incidence of lean MASLD in women between the ages of 50 and 59. In addition, less than 10% of women without obesity had abnormal glucose levels, highlighting the need for greater attention to MASLD in women with normal glucose levels compared to men [103].

Taken together, these findings highlight the pathogenetic complexity of the disease and the involvement of metabolic dysfunction. In addition, although the understanding of sex differences in the prevalence of MASLD in individuals with T2D and/or obesity remains unclear and conflicting, hormonal factors may significantly influence the development of MASLD and metabolic disorders, highlighting the need for further research to explore the complex relationship between weight, T2D, and MASLD, considering sex and age as key factors.

### 3.2. Cardiovascular Diseases

The strong association between MASLD and metabolic disorders places people at a significantly higher risk of mortality [2,68,92], with a 1.17-fold increased likelihood of CV disease compared to those without [104]. While CV disease has traditionally been thought of as more common in men and older adults [105,106,107], it follows a similar pattern to MASLD, with higher risks in men under 50 and increasing rates in postmenopausal women [5,60,108,109,110].

For individuals younger than 54 years, CVD causes total mortality of approximately 22% in men and 18.5% in women. After age 55, this trend reverses, with CVD mortality at 38.5% for men and 41% for women [85].

The PURE study, which examined gender differences in CV in people aged 35–70 years, found that women had fewer CV risk factors and lower rates of CV disease, likely due to greater use of primary prevention treatments (antiplatelet, antihypertensive, and lipid-lowering medications) and healthier lifestyles than men [111]. In addition, the protective effects of estrogens may delay the onset of CV disease [112], leading to the misconception that they are less common in women and therefore do not require as intensive preventive therapy as in men [113]. Studies have shown that women tend to experience CV events seven to ten years later than men [110,113,114,115], therefore the mean age of women (50 years) in the PURE study may be too early to detect significant CV risk effects. Furthermore, the higher pharmacologic burden observed in women for primary prevention [111] may reflect the worsening of traditional risk factors for MetS and MASLD, such as weight gain, insulin resistance, and hypertension, that begin in the peri-menopausal period. This suggests that the loss of the cardioprotective effects of estrogens [112,116] may contribute to the future development of CV disease. For example, after age 75, women account for the majority of cases of acute coronary syndrome (ACS), ST-elevation myocardial infarction, or non-ST-elevation myocardial infarction [115]. Moreover, in a sample of 63.245 people hospitalized for ACS, women were on average about 7 years older than men (73 vs. 66 years) and had a higher comorbidity burden [117]. The atypical symptoms of CV disease in women, which require early diagnosis for appropriate management, may also contribute to delayed diagnosis and treatment [112,118], as well as their underestimation in this population [105]. Endothelial function begins to decline early in menopause, often before signs of subclinical atherosclerosis are evident. This change may play a role in the development of “unexplained” chest pain and shortness of breath, which are often attributed to “stress” or “menopausal symptoms. However, women who experience these symptoms are twice as likely to develop ischemic heart disease over the next 5–7 years [119]. Notably, women were also less likely to undergo cardiac investigations [111,116] and received secondary pharmacotherapy less frequently [111], probably due to a higher likelihood of experiencing medication side effects [111,116], contributing to the lack of diagnosis and treatment.

MASLD includes cardiometabolic risk factors that develop in women primarily during menopause [120]. While each factor independently affects CV risk, their combined effect significantly increases the risk, contributing to a 1.5-fold increase in all-cause mortality [121]. This highlights the need for screening, prevention, and treatment of these risk factors, such as weight gain, dyslipidemia, hypertension, and T2D, in women during the menopausal transition [85,119].

The presence of hepatic steatosis in women with MASLD reduces the protective effects of estrogen, leading to higher rates of CV disease [30,122,123]. For example, in a Swedish cohort of 10,422 adults with biopsy-confirmed MASLD, women showed a stronger positive association between MASLD and the incidence of major adverse CV events (MACE) than men [124]. This increased mortality has been linked to the increasing diagnosis of MASLD in midlife, as well as the recognition of age as a risk factor for the fibrotic and cirrhotic progression of MASLD in women, largely due to the loss of estrogen [122]. Even after adjustment for several risk factors, the Fibrosis-4 Index (FIB-4) score [125], which has been demonstrated to have superior performance in detecting advanced stages of fibrosis among non-invasive liver disease assessments (NILDAs) [126], was associated with a higher risk of ischemic stroke in women but not in men [125]. These sex differences may be influenced by the combined effect of MASLD and menopausal hormonal changes that increase CV risk [127,128]. A study of 41,005 MASH patients found that men had higher rates of most CV risk factors and diseases than women. However, despite the relatively young average age of the study cohort (61 years), women had higher rates of obesity and T2D, similar stroke rates to men, and a higher prevalence of MASH [129]. This suggests that the presence of a dysmetabolic state in women with MASLD reduces the protective effects typically associated with female sex, making women more susceptible to CV disease [94]. In addition, the excess of VAT, which is common in menopause and MASLD, may be more harmful in women than in men. For example, the area of VAT, measured by computed tomography (CT) scan, has been found to be independently associated with heart failure with preserved ejection fraction, hemodynamic abnormalities [130], and increases in left ventricular mass (LVM) and mean myocardial wall thickness (MMWT) in women, but not in men [131]. Furthermore, in advanced menopausal women, the positive multiplicative interaction between different measures of abdominal obesity (WC, BMI, WC+BMI) and MASLD increases the cumulative incidence and risk of CV disease, underscoring the need to monitor excess adiposity to reduce cardiovascular risk in these individuals, especially in women with elevated BMI and WC [104]. As a result, current clinical risk assessments may underestimate the CV risk in women [123], emphasizing the importance of understanding sex-specific risk factors and early screening for women with MASLD to develop targeted prevention strategies.

### 3.3. Depression, Anxiety, and Neurodegenerative Diseases

Insulin resistance, inflammation, and oxidative stress are key mechanisms in the development of obesity, T2D, MetS, and MASLD. These factors also contribute to cognitive decline, particularly in the aging population [132,133,134,135]. Central nervous system (CNS) disorders, including neurodegenerative diseases and mood disorders (depression and anxiety), are common in MASLD [134]. This effect is particularly evident when liver disease is accompanied by glucose metabolism abnormalities [136], and advanced progression (fibrosis and MASH) [132,136]. Insulin signaling is crucial for synaptic plasticity, neuroprotection, and neuronal growth [137], which helps explain how insulin resistance, driven by increased VAT and the release of pro-inflammatory cytokines, impairs cognitive function in MASLD [138]. The discovery of central insulin resistance in Alzheimer’s disease has made it a potential therapeutic target [137]. While VAT worsens the relationship between aging and cognitive decline in both men and women, estradiol appears to mitigate this effect in women [139,140]. By regulating metabolism, improving insulin sensitivity, and preventing inflammation [141], estrogens offer neuroprotection against cognitive decline in middle-aged women [140], explaining the association between cognitive impairment and MASLD is stronger in postmenopausal women [141,142,143,144,145,146]. In women over 50, the loss of estrogen’s protective effects [141], associated with higher prevalence of MASLD and fibrosis [37,146,147], strengthens the pathophysiological link between MASD and cognitive decline [141,148,149,150,151].

As a result, MASLD (as assessed by the hepatic steatosis index (HSI) and FIB-4 scores) was associated with greater beta-amyloid deposition in postmenopausal women, but not in men [141], and a higher risk of dementia, particularly in people aged 45–54 years and those with T2D [152]. Middle-aged women with MASLD also have a higher prevalence of depression, which worsens with the severity of steatosis [143,144,153]. On the other hand, anxiety and depression are associated with a higher predisposition to MASLD [153], and worse treatment outcomes [154]. In fact, these mental health conditions are associated with higher levels of binge eating disorders (BED) and emotional eating [155,156], especially in postmenopausal women [157], that negatively affect diet, weight [156], and quality of life (QoL), creating barriers to weight loss and effective MASLD treatment [154]. Poor diet can also disrupt the gut microbiota, contributing to fat deposition in the liver [134,158] and cognitive decline in MASLD [133]. Overall, anxiety and depression may increase the risk of future metabolic disorders, suggesting a bidirectional relationship [153] that warrants further attention, especially after menopause.

## 4. Sex Differences in the Pathogenesis of MASLD: Should the Focus Be on Postmenopausal Women?

The new MASLD nomenclature emphasizes the strong association between hepatic steatosis and metabolic abnormalities characteristic of MetS, which increase the risk of progression to more severe liver disease [91]. Initially, the “two-hit” theory described the pathogenesis of MASLD, where the first hit is liver damage caused by TG accumulation due to insulin resistance and an obesogenic lifestyle, which makes the liver more susceptible to a “second hit” from inflammatory cytokines, adipokines, mitochondrial dysfunction, and oxidative stress. These factors induce inflammation and fibrosis in hepatocytes [159]. The “multiple hit” hypothesis extends this theory by suggesting that in genetically predisposed individuals, a combination of factors such as insulin resistance, adipose tissue dysfunction, sex hormones, diet, gut microbiota, and genetic/epigenetic influences contribute to hepatic steatosis and inflammation [160].

As has been widely reported, the prevalence of MASLD, along with hepatic and extrahepatic disease, follows a similar pattern. Women typically have a lower risk of hepatic, cardiometabolic, and cognitive disease during their reproductive years, but lose this protection after menopause, resulting in a disease prevalence comparable to or higher than that of men [36,60]. The hormonal changes that occur after menopause, especially the loss of estrogen, are thought to contribute to this pattern in women [5]. However, the multiple factors outlined in the multiple-hit hypothesis amplify these hormonal shifts and influence the risk profiles and phenotypes of postmenopausal women [12,36], exacerbating the risk of developing MASLD and its progression (Figure 1).

Given that MASLD is the leading cause of liver transplantation in women without HCC, but women are less likely to receive a transplant due to the severity of the disease, exploring the interaction between these factors and the pathogenesis of MASLD may help explain changes in prevalence after menopause and guide targeted therapeutic strategies for women, focusing on this subgroup.

### 4.1. Sex Differences in Genetic Predispositions

Metabolic diseases such as obesity and T2D show familial clustering, suggesting a strong hereditary component to the risk of developing MASLD and related liver diseases [161,162]. Because genetic predisposition differs between men and women, affecting disease susceptibility and progression [163], the identification of genetic variants, especially with the advent of precision medicine, can help develop targeted treatments. When combined with clinical factors, these genetic insights can help predict the risk of progression to MASH, cirrhosis and HCC [2].

Recent genome-wide association studies (GWAS) have identified important single nucleotide polymorphisms (SNPs) associated with susceptibility to MASLD. These SNPs are involved in the regulation of glucose and lipid homeostasis [164], with some showing sex-specific associations [165]. Key genetic variants, such as PNPLA3 p.I148M and TM6SF2 p.E167K, contribute to TG accumulation in the liver [159,164], increasing the risk of MASH, fibrosis, cirrhosis, and HCC [49]. The presence of PNPLA3 variants has been shown to increase the likelihood of progression to MASH in women, but not in men. In men, it has been shown to increase the likelihood of progression to fibrosis and liver dysfunction only in the later stages of chronic liver disease, particularly when associated with alcohol consumption [163]. A study of a large European cohort of biopsy-confirmed MASLD patients found that the PNPLA3 variant influenced liver-related events in non-obese women aged 50 years or older, suggesting a need to focus on this subgroup [166]. Although the TM6SF2 variant affects the development and severity of SLD similarly in both sexes, a stronger interaction between female sex and the PNPLA3 p.I148M variant was observed in postmenopausal women (≥55 years) with relatively higher estradiol (E2) levels. In carriers of the p.I148M variant, persistently higher E2 levels due to estrone conversion in hepatocytes may synergize with inflammation and insulin resistance, resulting in induction and accumulation of the variant protein, leading to hepatic steatosis [50]. Thus, downregulation of PNPLA3 may be more effective in women, with postmenopausal women carrying the variant potentially representing a high-risk subgroup for targeted treatments. This may explain the higher risk observed in these women, although predicting individual disease remains challenging [2]. However, other factors may contribute to the greater hepatic expression of PNPLA3 in women compared to men [50].

GWAS have identified loci associated with fat distribution, showing a stronger association of key SNPs in women compared to men [167,168,169], and suggesting that adiposity traits are more heritable in women [169]. The sex chromosome complement may also contribute to the sexual dimorphism observed in metabolic traits [22,167]. For example, mice with XX chromosomes have up to a two-fold increase in obesity [170,171], weight gain, elevated lipid and insulin levels, and fatty liver when fed a high-fat diet (HFD) [170]. Similar patterns are observed in syndromes with an extra X chromosome, such as Klinefelter syndrome (XXY), where affected males have higher liver function markers, increased body fat, abdominal fat, TG, cholesterol, and insulin resistance, suggesting the involvement of MetS components [22,172,173]. In contrast, women with Turner syndrome (partial or complete loss of the X chromosome) have a variety of liver problems, including steatosis, steatohepatitis, cirrhosis, and increased weight and BMI, probably due to estrogen deficiency [174,175]. In humans, it remains difficult to distinguish the roles of sex chromosome complement and gonadal hormones. This complex network of genetic imprinting and gonadal hormones is influenced by multiple factors. Understanding these interactions could help identify potential therapeutic targets and enable treatments tailored to an individual’s sex profile [102], especially for women.

### 4.2. Estrogens in Glucose and Lipid Metabolism

The influence of age and gender on the clinical features of MASLD/MASH is well documented [147], with sex hormones playing a critical role in regulating metabolic processes in the liver [22]. Estrogens help prevent MASLD by directly exerting their metabolic effects on hepatocytes and indirectly improving insulin resistance and dyslipidemia through the modulation of fat distribution [7]. During menopause, estrogen deficiency leads to altered glucose and lipid metabolism, creating a dysmetabolic environment that promotes hepatic steatosis and progression to hepatic inflammation and fibrosis [5,15]. 17β-estradiol (E2) is the primary female sex hormone during the reproductive years and plays a critical role in regulating interactions between the liver and adipose tissue [176]. In the female liver, under normal physiological conditions, estrogen signaling suppresses the expression of genes involved in lipogenesis, such as fatty acid synthase (FAS), acetyl-CoA carboxylase (ACC), stearoyl-CoA desaturase (SCD), and sterol regulatory element-binding protein 1 (SREBP1), while promoting the expression of genes involved in fat oxidation [6,177], such as carnitine palmitoyltransferase (CPT1) [6]. Increased mitochondrial fatty acid oxidation and decreased de novo lipogenesis (DNL) protect hepatocytes from fat overload [5], a key pathogenic mechanism in the development of MASLD and lipotoxicity. In addition, estrogens influence the maintenance of hepatic cholesterol balance by stimulating lipoprotein synthesis and very low-density lipoprotein (VLDL) particle release, increasing HDL production, and removing oxidized low density lipoprotein [109]. The enhancement of cholesterol absorption in the liver and reverse cholesterol transport by estrogen receptor alpha (ERα) may explain the increase in cholesterol levels from premenopausal to postmenopausal status, which contributes to the higher prevalence of MASLD in postmenopausal women [60,177]. For example, ovariectomy in female rats leads to the development of MASLD/MASH, as it increases SREBP1 expression and decreases genes responsible for VLDL secretion, resulting in intrahepatic TG accumulation and elevated ALT levels. These changes further exacerbate hepatic fibrosis and may drive the progression of MASH [178].

The innate immune response to damaged hepatocytes plays a key role in the development of MASH, as the recruitment of inflammatory immune cells contributes to liver inflammation during the progression of MASLD [179]. Estrogen helps mitigate liver damage once it has occurred [180] by regulating inflammatory and fibrotic pathways in Kupffer Cells (KCs) and HSCs [5,180]. As the predominant hepatic immune cell type, KCs are essential for activating the inflammatory response, recruiting immune cells, and promoting insulin resistance, free fatty acid (FFA) accumulation, and inflammation, all of which contribute to fibrosis development [179]. This process activates key transcription factors such as c-Jun N-terminal kinase (JNK) and κ light chain enhancer of B cell (NF-κB) [41,181], which trigger the release of pro-inflammatory cytokines such as tumor necrosis factor (TNFα), interleukin-16 (IL-16), and interleukin-1β (IL-1β), exacerbating insulin resistance and chronic inflammation [41]. Chronic liver injury also activates HSCs, which promote excessive accumulation of extracellular matrix proteins, primarily collagen, leading to fibrosis [182]. Although E2 helps mitigate fibrogenesis by inhibiting these pro-inflammatory pathways [41], research has shown that preventive and therapeutic treatments with 17α-E2 can slow or even reverse the progression of liver fibrosis by significantly inhibiting collagen production in chronic liver injury. As a result, it is believed that E2 likely exerts its effects through multiple mechanisms involving different cell types to help alleviate liver injury and fibrosis [182].

The improvement in mitochondrial quality and biogenesis regulated by peroxisome proliferator-activated receptor γ coactivator 1 alpha (PGC-1α) is also estrogen-dependent [183]. Through the ERα/Sirtuin1/PCG-1α signaling pathway, E2 improves mitochondrial quality and biogenesis regulated by PGC-1α, reduces mitochondrial oxidative stress caused by lipotoxicity, and interferes with JNK signaling, thereby limiting the onset of insulin resistance [41,184,185]. Estrogen signaling also protects against HCC by inhibiting inflammasome activation and regulating NF-κB pathways, shifting KCs from a pro-inflammatory to an anti-inflammatory phenotype, with this effect being more pronounced in women [186] than men [102]. The anti-inflammatory effects of estrogens may help explain why premenopausal women, who have a lower prevalence of MASLD, tend to have lower CRP levels than postmenopausal women, who have a higher inflammatory status [187]. Overall, the protective effects of estrogen against liver injury and fibrosis help explain the increased risk of advanced fibrosis in postmenopausal women [180,188].

At the mechanistic level, estrogens regulate liver lipid metabolism through their receptors, primarily ERα, which is expressed in hepatocytes and KCs, while ERβ is expressed in HSCs [5,6,102]. Mouse models have shown that ERα signaling promotes metabolic flexibility in women and protects against obesity and metabolic dysfunction induced by a HFD [189], such as fatty liver, elevated circulating lipids, ectopic fat accumulation, impaired glucose tolerance, and insulin resistance, common in MetS and menopause [190]. In fact, although male mice develop obesity and metabolic dysfunction more rapidly than females when exposed to a HFD, females experience age-related changes in liver function, including altered fatty acid oxidation during perimenopause, leading to increased hepatic fat accumulation [40]. These changes are particularly common in postmenopausal women [189,190]. For example, ovariectomy in female mice results in overexpression of genes involved in lipid accumulation in the liver, and male genes only in those with ERα. This suggests that estrogen deprivation reprograms the liver to a male-like profile, contributing to MASLD in postmenopausal women [191]. Liver-specific ablation of ERα in female LERKO mice results in increased DNL, elevated circulating lipids, and ectopic fat accumulation [177,189], further supporting the idea that the loss of estrogen protection after menopause increases the prevalence and severity of MASLD, including advanced fibrosis and MASH [26,180]. Finally, ERα has been shown to accelerate hepatic steatosis, M1 macrophage infiltration, and fibrogenesis in obese female mice [192]. Overall, this highlights hepatic ERα as a potential target for the treatment of postmenopausal MASLD.

Estrogen plays a key role in glucose metabolism by stimulating glucose uptake in skeletal muscle and adipose tissue [193]. However, in mice lacking estrogen signaling (global or hepatic ERα knockout), this disruption leads to hepatic insulin resistance and increased hepatic glucose production (HGP), resulting in hyperglycemia [194]. This highlights the essential role of ERα in the regulation of glucose and energy balance. Estrogens suppress HGP and improve insulin sensitivity via the ERα-phosphoinositide 3-kinase-Akt-Foxo1 pathway [41,194], with Foxo1 regulating glucose-6-phosphatase (G6pc), a critical enzyme in HGP [195].

In addition, ERα improves hepatic insulin sensitivity by regulating insulin receptor substrate 1 (IRS-1) [194], and glucose transporter 2 (GLUT-2), promoting glycogen synthesis and insulin release [26]. Although regulation of HGP plays a more important role in reducing insulin resistance than muscle glucose uptake [194,196], enhancing muscle glucose utilization also helps improve insulin sensitivity [193]. Estrogens also support β-cell function by inhibiting glucotoxicity-induced apoptosis and reducing inflammation [193].

Overall, this explains why premenopausal women generally have better glucose tolerance, insulin sensitivity, and glucose utilization than men and postmenopausal women. In the latter group, estrogen deficiency leads to dysglycemia and reduced hepatic insulin clearance [177,194]. These changes in glucose metabolism secretion during menopause are also associated with β-cell apoptosis, decreased insulin secretion, and increased hepatic degradation [85].

These findings support the need to consider the role of estrogens in regulating glucose and lipid metabolism as they influence susceptibility to multisystemic metabolic diseases such as MASLD.

### 4.3. Involvement of Estrogens in the Adipose Tissue Dysfunction After Menopause

Excessive energy intake, along with hormonal, behavioral, and metabolic factors, contributes to obesity and increased fat storage [7]. Obesity is a major risk factor for MASLD, with fat type and distribution being more important than total body fat [102]. Adipose tissue distribution, which affects cardiometabolic risk independent of body weight or fat percentage, is significantly influenced by sex hormones [60]. Estrogen promotes fat storage in the gluteal–femoral region in women (pear-shaped obesity), while testosterone causes visceral fat accumulation in men (apple-shaped obesity) [7,102].

Gluteo–femoral SAT stores fat more efficiently, enhances fat browning, and promotes the production of anti-inflammatory adipokines, offering protection against MetS and MASLD [176]. Its large adipocytes and ability to expand through hyperplasia and hypertrophy act as a “metabolic reservoir”, safely storing excess energy. This prevents harmful abdominal and intra-abdominal fat accumulation and avoids lipotoxicity in insulin-sensitive tissues. In addition, the higher activity of lipoprotein lipase (LPL) in subcutaneous adipocytes of the lower body in women suggests greater TG clearance compared to visceral adipocytes in men with obesity [171]. Estrogens also regulate genes involved in brown adipose tissue (BAT) function, such as uncoupling protein 1 (UCP-1), which is critical for energy balance, and promotes the “browning” process of white adipocytes [197]. This process, along with adipocyte hyperplasia, helps protect against hypoxia and inflammation in adipocytes [176,198]. Furthermore, overexpression of ERα has been shown to improve mitochondrial function in hepatocytes, reduce hepatic lipid accumulation, and protect against hepatic insulin resistance and steatosis in mice [199].

On the contrary, excess VAT is characterized by dysfunctional adipocytes resulting from hypertrophic expansion and mitochondrial dysfunction, leading to a pro-inflammatory state and insulin resistance [171]. The lack of anti-lipolytic action of insulin makes VAT more susceptible to lipolysis, leading to an increase in circulating FFAs that are redistributed to organs, including the liver, exacerbating insulin resistance, inflammation, and metabolic stress in hepatocytes [25,176].

The metabolic differences associated with the effects of sex hormones on fat distribution are reflected in the correlation between changes in estrogen levels and fat percentage over a woman’s lifetime [60,171]. After menopause, the decline in estrogen promotes a redistribution of lipids from SAT to a more central, android fat distribution [60,177,200]. This helps explain the favorable cardiometabolic profile seen in premenopausal women, who have better glucose regulation, increased insulin sensitivity, and a less atherogenic plasma lipid profile [171]. In contrast, as happens in postmenopausal women [85], people with higher VAT mass tend to be more insulin resistant, have impaired glucose metabolism, and are more likely to develop MASLD [15,60]. Similarly, research indicates that young women with reproductive dysfunction or those who have undergone ovariectomy experience a higher prevalence of MASLD compared to their fertile counterparts [50].

According to research, menopause alters the phenotype of adipose tissue in both SAT and VAT, reducing the number of adipocytes [200]. This may suggest that estrogens may also regulate cell apoptosis [201]. Additionally, hypertrophic changes in VAT were associated with reduced insulin sensitivity [200] and increased peroxisome proliferator-activated receptor gamma (PPAR-γ) expression, while decreasing FASN expression in SAT. These changes are thought to be an adaptive protective response to lipid accumulation [60,200].

It has also been suggested that E2 may centrally regulate thermogenic adipose tissue by activating hypothalamic AMP-activated protein kinase (AMPK), a pathway that acts as an energetic sensor of nutritional status and cellular metabolism [202,203]. This also supports the role of sex hormones in influencing the higher levels and activity of BAT in women compared to men, attributed to higher thermogenic proteins such as UCP1 and PGC1α [204,205]. A decrease in oxidative metabolism and glucose uptake in BAT has been observed in postmenopausal women compared to premenopausal women, suggesting a loss of thermogenic function associated with reduced E2 levels [206]. Furthermore, ovariectomy has been shown to affect several proteins involved in the metabolic pathways of BAT [207].

Given the role of ERα in the regulation of mitochondrial biogenesis and oxidative phosphorylation through the induction of PGC1 and CPT-1, as well as lipid export from the liver via the secretion of TG-rich VLDLs [31,41,208], the reduced oxidative capacity of the liver during menopause shifts hepatic lipid metabolism from fatty acid oxidation to lipogenesis. This leads to excessive hepatic fat accumulation, lipotoxicity, and chronic inflammation [31,60,209].

In postmenopausal women, VAT accumulation has also been associated with increased inflammation in SAT, with higher levels of hypoxia-inducible factor (HIF-1α) in both adipose tissues, which mediates the hypoxic-inflammatory response associated with hypertrophic adipocyte expansion. This factor contains an estrogen response element in its promoter, suggesting that estrogen signaling through HIF-1α may help prevent hypoxia and fibrosis [200]. In fact, adipose tissue also functions as an endocrine organ, releasing several proteins that regulate metabolism, inflammation, and fibrosis [12]. In particular, visceral obesity affects the liver by causing an imbalance in the secretion of these adipokines from dysfunctional adipose tissue, exacerbating the cycle of insulin resistance, hepatic steatosis, inflammation, and fibrosis [41]. Two key adipokines, adiponectin and leptin, show sex differences in their levels [7,179], with women typically having higher levels than men [179,210]. People with MASLD/MASH have altered adipokine levels, with lower adiponectin and higher leptin [211,212]. Adiponectin improves insulin sensitivity, reduces inflammation and fibrosis [12], and offers cardioprotective, anti-obesity, and hepatoprotective effects [7,211]. Its synthesis is stimulated by E2, helping to protect the liver and cardiovascular system before menopause, while reducing its levels after menopause [7]. Adiponectin levels also decrease as MASLD progresses to MASH [175], with this relationship being particularly pronounced in women [175]. Leptin, on the other hand, is known as the “starvation hormone”, which is synthesized by adipose tissue and acts as a metabolic regulator that can reduce food intake [179,213]. By improving insulin sensitivity and promoting lipid mobilization, leptin helps reduce fat accumulation in the liver, thereby protecting against MASLD [214,215]. Physiologically, women appear to be more sensitive to the anorexigenic effects of leptin than men, probably due to the sensitizing effects of estrogen [214,216]. However, in MASLD, leptin shows dose-dependent activity, becoming inflammatory and fibrotic at higher levels [175]. For example, women with obesity with high levels of leptin develop resistance to its effects rather than sensitivity [214], showing a positive correlation with serum insulin levels, insulin resistance, and liver damage, including both steatosis and inflammation, contributing to MASLD/MASH [214,215]. This explains why adiponectin levels decrease in liver disease while leptin levels remain elevated [7,211].

The adiponectin/leptin (A/L) ratio has been suggested as a marker of adipose tissue dysfunction, also known as secretory adiposopathy, characterized by altered lipid metabolism [217], and a more accurate index for steatosis lesions and their progression to MASH [218]. The abnormal expression of these adipokines has also been associated with the development and prognosis of HCC, suggesting them as potential biomarkers for the early diagnosis and progression of liver cancer [219].

Taken together, this suggests the involvement of sex steroids in the regulation of numerous processes in adipocyte function, including their endocrine activity, which is critical for maintaining normal metabolic functions. Therefore, the loss of estrogen after menopause may affect the metabolic and secretory function of adipocytes, increasing susceptibility to various metabolic diseases, including MASLD, and thus abolishing the sex-specific incidence of this pathology [177].

### 4.4. Gut Microbiome Changes After Menopause

The gut microbiome plays a critical role in regulating metabolic processes through the gut–liver axis, and its influence on metabolic diseases is increasingly recognized [220]. Dysbiosis, or changes in the composition of the gut microbiota, has been associated with all stages of MASLD [211,221], including steatosis, MASH, advanced fibrosis, cirrhosis, and HCC [222].

The influence of the gut microbiome on MASLD involves complex microbiome–host interactions. Increased intestinal permeability, or “leaky gut,” leads to the release of microbiota-derived metabolites that are recognized as biomarkers or therapeutic targets for MASLD [211,223]. For example, metabolites such as lipopolysaccharide (LPS, a major component of the outer membrane of Gram-negative bacteria) or trimethylamine N-oxide, contribute to systemic inflammation, alterations in bile acid and choline metabolism, and the production of short-chain fatty acids (SFCA) and ethanol [22,25,224,225].

Emerging evidence indicates that sex differences significantly shape the gut microbiome. In a rat model of HFD-induced obesity, HFD decreased microbial taxon diversity, with female rats having even lower diversity than male rats. Specifically, male HFD rats had higher levels of Firmicutes and Bilophila, bacteria that disrupt intestinal barrier function and contribute to hepatic steatosis. In contrast, females had higher levels of propionate, an SFCA beneficial for obesity and metabolic disease. These findings suggest that female rats may experience less severe liver inflammation and steatosis than males [226]. Another MASLD model showed differences in microbiome composition between male and female rats, with greater diversity in females. In addition, male rats showed differential expression of Toll-like receptor 4 in the liver, which correlated with their different microbiome profiles compared to females [227]. These microbiome differences also extended to gut inflammation, highlighting the role of sex in shaping microbiome-associated metabolic changes [228].

It is reasonable to assume that sexual dimorphism in the gut microbiome is influenced by sex hormones [228,229,230] and that this interaction plays an important role in the development of metabolic diseases, particularly affecting key aspects such as intestinal barrier integrity and inflammatory status [231].

This helps explain why, similar to ovariectomized mice with reduced estrogen levels, the changes in the gut microbiota observed during menopause [230] lead to the neutralization of sex differences in microbiota composition and function [232]. Serum estradiol has been associated with greater gut microbiota diversity in premenopausal women, as well as a greater ability to maintain the gut barrier and protect against gut injury [233]. After menopause, women’s gut microbiomes become more similar to men’s in both composition and function, suggesting a masculinization of the microbiota [234]. In fact, the decline of female hormones during menopause causes shifts in the gut microbiome, with certain species being associated with cardiometabolic profiles characteristic of this stage of life, increasing the risk of MetS. Specifically, the increase in VAT is associated with changes in gut microbiota populations and markers of gut inflammation. For example, higher VAT correlates with a lower Firmicutes/Bacteroides ratio, increased abundance of gut proteobacteria, and markers of metabolic endotoxemia such as LPS. In contrast, lower VAT is associated with SCFA-producing microbes [235]. Further studies with larger populations are needed to identify patterns in taxa affected by menopause [233].

Studies have also found that microbial profiles differ between men and women, with sex-specific variations in MASLD patients. For example, amino acid degradation pathways and production of harmful metabolites were particularly elevated in middle-aged women with MASLD, and only three genera differed between MASLD and control groups in both sexes. This suggests that the disease affects the composition of the microbiota in a sex-specific manner [236].

Given that menopause-associated changes in the gut microbiome may be associated with adverse cardiometabolic risk, including the development of MASLD in postmenopausal women, understanding these microbial variations may reveal sex-specific or personalized pharmacotherapies for MASLD [102].

## 5. Applicability of Non-Invasive Diagnostic Tools for Screening for Postmenopausal Liver Disease

Given the clinical burden of MASLD, non-invasive tests [237] offer early detection of liver disease, cost-effectiveness in the management of metabolic dysfunction, and overcoming the limitations of invasiveness of liver biopsy, the gold standard for assessing liver fibrosis [238]. Because liver fibrosis can regress with effective treatment of the underlying disease, especially in its early stages, non-invasive screening tools are particularly valuable for patients with risk factors such as obesity and diabetes [239]. Given the increasing prevalence of MetS, MASLD, and advanced fibrosis, the use of non-invasive diagnostic tools to screen for liver disease is crucial in postmenopausal women.

A significant increase in VAI, a marker of dysfunctional abdominal fat, has been observed in postmenopausal women and its association with MASLD reflects the metabolic shift associated with reduced estrogen levels [108]. The decline in estrogen may also explain the strong correlation between age and the non-invasive FLI index as a key predictor of hepatic steatosis and MASLD in middle-aged women. It highlights the need to revise FLI cut-off values for women over 50 to better reflect the association between MASLD and metabolic factors that affect women more than men, particularly with regard to liver outcomes [240]. In addition, a low A/L ratio has been found in middle-aged women with higher BMI and adiposity, particularly those with larger abdominal and omental adipocytes. These women have higher rates of lipolysis and reduced expression of GLUT-4 in adipose tissue, a marker of insulin resistance [217].

Transient elastography (TE) is also a reliable and non-invasive alternative to liver biopsy that can quantify liver stiffness as a marker of fibrosis better than simple blood tests [238]. Given the higher susceptibility to liver fibrosis in women after menopause, this imaging diagnostic tool may be more reliable in this subgroup screening. For example, TE has shown an independent association between female visceral obesity and more severe hepatic steatosis in postmenopausal women, even after adjustment for HOMA-IR and T2D. The shift of fat to the visceral area after menopause makes visceral obesity more susceptible to fatty liver compared to premenopausal women and men, confirming the impact of menopause on liver histology in addition to sex [241]. A cross-sectional study further supported this by showing a significant association between abdominal obesity and MASLD as assessed by TE in middle-aged women [242]. The cost-effectiveness of non-invasive liver fibrosis screening with TE has been validated in middle-aged people from the general population and in patients with risk factors, half of whom were women [239]. Particularly, TE has identified a high prevalence of significant fibrosis associated with metabolic risk factors typical of MetS in a large cohort of middle-aged individuals, nearly 60% of whom were women [243].

In light of these findings, further research is needed to explore how non-invasive screening tools, including NILDAs or imaging techniques such as TE, could serve as an effective public health strategy to prevent MASLD and its progression in postmenopausal women. Increased awareness and early identification of women at risk would facilitate the timely implementation of treatments for underlying metabolic conditions, potentially reversing the increasing trend of MASLD progression after menopause.

## 6. Non-Pharmacological Treatment of MASLD: How Postmenopausal Hormonal Fluctuations Affect Women’s Dietary Behavior

Lifestyle interventions, such as diet and physical activity, are essential for managing and preventing the metabolic changes associated with MetS and MASLD. There is a direct correlation between the amount of weight loss achieved by a low-calorie diet and the extent of improvement in biomarkers of liver injury. However, improving diet quality by avoiding ultra-processed foods (high in sugar and saturated fat) is critical to preventing MASLD and improving liver-related clinical outcomes [2]. The Mediterranean diet, rich in olive oil, vegetables, fruits, legumes, fish, and whole grains, has been associated with a reduced risk of MASLD [2,244,245], while a Western diet high in red meat, sugar, and saturated fat increases the risk of MASLD [2,244,246], also affecting the microbiome, intestinal barrier, bacterial translocation, and neuroinflammation [134,158].

Diet quality appears to have a greater effect on VAT in women than in men [247], emphasizing the need to consider gender differences in dietary patterns and their impact on diet quality and MASLD risk.

Middle-aged women are more likely than men to consume healthier foods, such as fruits and vegetables [248,249], and are more likely to follow the Mediterranean diet [250,251]. Adopting these healthy eating habits from an early age [252] may partly explain their lower risk of premenopausal MASLD. However, the “menopausal transition” is a period characterized by both physical and psychological changes [253]. Hormonal fluctuations increase the risk of obesity, especially central obesity, which in turn contributes to inflammation and cardiometabolic diseases such as MASLD [254]. In addition, these fluctuations lead to depression and decreased physical activity, further exacerbating weight gain. This explains the higher prevalence of BED in women, with the female-to-male ratio ranging from 2:1 to 10:1 [255,256], especially during menopause [255,257]. BED is associated with several comorbidities, including MetS, sleep disturbance, CV problems, somatic symptoms, and reduced QoL in older women [258]. Sleep and mood disorders, such as depression, anxiety, and emotional distress [255,259], combined with somato-vegetative symptoms (hot flashes, palpitations, insomnia), which are common during menopause, contribute to weight gain and an increase in visceral fat mass [260]. This creates a vicious cycle that helps explain the rise in obesity during menopause.

The primary biological factor influencing gender differences in eating behavior is the alteration of gonadal hormones [259,261], which affect appetite, caloric needs, and energy balance [250,262,263]. E2 plays a critical role in regulating energy expenditure, metabolism, and body composition by modulating food intake [253]. Activation of the ERα receptor reduces appetite by increasing anorexigenic peptide expression in POMC neurons and decreasing orexigenic peptide expression in NPY/AgRP neurons [264], with female-specific effects, particularly in the hypothalamic arcuate nucleus (ARC) and nucleus of the solitary tract (NTS) [265]. E2 also interacts with other hormones [171,266], such as ghrelin (an orexigenic gastric hormone), leptin, cholecystokinin, and GLP-1 to promote the cessation of food intake [264,267,268]. It also reduces food-related reward in the ventral tegmental area (VTA) [264]. Hormonal fluctuations throughout the menstrual cycle influence emotional eating, with food intake peaking during the luteal phase (lower estradiol levels) and decreasing during the ovulatory phase (higher estradiol levels) [259,261]. A reduction in E2 levels may worsen uncontrolled eating disorders, such as BED [255,259,263,264,269], as evidenced by the increased risk of compulsive overeating in adult female rats following ovariectomy [270], and in postmenopausal women [268]. In addition, the absence of ERα in the liver may impair the nutritional response, especially under compromised nutritional conditions [177].

Estrogen deficiency during menopause disrupts circadian rhythms and sleep patterns [271], along with hormones that regulate appetite and body composition. Specifically, women experience a more pronounced decrease in leptin levels compared to men after sleep deprivation, which leads to increased food intake and lipid accumulation in adipocytes [272]. Chronic disruption of circadian rhythms, along with increased nocturnal snacking, is a known risk factor for metabolic disease, as evening or nighttime caloric consumption is associated with a higher risk of hepatic steatosis [273]. In fact, a positive association has been found between severe vasomotor symptoms in postmenopausal women and the presence of MASLD [274]. Notably, these symptoms are often associated with lower adherence to the Mediterranean diet [260], and increased consumption of highly processed foods, saturated fats, and sugars [275,276].

Social and environmental stressors significantly affect the relationship between ovarian hormones and food preferences, with emotional stress often leading to increased consumption of fatty and sugary foods [277]. Women in particular experience a strong link between body dissatisfaction, shaped by socio-cultural ideals, and disordered eating, beginning in adolescence [278,279]. Higher rates of eating disorders in women, coupled with their greater likelihood of experiencing anxiety and mood disorders [255,262], are linked to increased concerns about body shape and weight, particularly in perimenopausal women compared to premenopausal women [255,280].

Hormonal changes, combined with midlife lifestyle factors, can lead to body changes and dissatisfaction [281], as women move away from the ideal of female beauty [282]. This concern about physical appearance often triggers negative thoughts and anxiety, which can lead to the development of unhealthy eating behaviors [283]. It has been reported that women tend to experience hunger spikes in the afternoon, often accompanied by cravings [248]. These food cravings are associated with episodes of uncontrolled eating [284], which reflect emotional rather than physical hunger, encouraging the consumption of high-calorie foods [262]. This helps explain why middle-aged women often experience BED despite adopting healthier eating habits [248,285].

Research suggests that postmenopausal women with MetS have an altered dietary profile, with higher intakes of refined carbohydrates, sweets, and fats, which is reflected in changes in anthropometric and lipid parameters [286]. Furthermore, among middle-aged individuals with at least two typical features of MetS, women were found to be more susceptible than men to the metabolic effects of the glycemic index of their diet. Given the high prevalence of MetS and MASLD after menopause, the negative impact of this life stage on women’s body composition and glucose homeostasis contributes significantly to the higher incidence of metabolic disorders [287]. An increase in carbohydrate consumption, especially simple sugars, promotes DNL, and fructose added to sugary beverages has been identified as a key factor in the development of MASLD and MetS [288].

Although human clinical data are limited, preclinical studies suggest sex differences in response to these dietary patterns [12]. For example, a fructose-rich diet, but not caloric excess, leads to small adipocyte proliferation, VAT inflammation, and the development of insulin resistance in female rats but not in males [289]. In a mouse model of fructose supplementation with a HFD, females had significantly higher MASH (steatosis, inflammation, and fibrosis) scores compared to males [290], along with a marked increase in hepatic TG [291]. In contrast, males showed only a modest increase in hepatic TG levels and no evidence of hepatic steatosis [292]. Despite fructose-mediated activation of carbohydrate response element binding protein (ChREBP) and DNL in both sexes, males appeared to counterbalance lipid accumulation in the liver through increased β-oxidation [292], while females showed reduced fat catabolism [291]. Moreover, in males, increased leptin levels correlated positively with β-oxidation [292], suggesting that it serves as a protective mechanism against lipid accumulation. Therefore, while MASLD may develop earlier in men, the severity of the disease is often greater in women.

Overall, the interaction of hormonal and social factors is a significant risk factor for the development of unhealthy eating habits after menopause, which are closely linked to obesity and depression [157,258], and the presence of typical MetS criteria [258]. These factors negatively impact QoL, create barriers to weight loss, and interfere with achieving treatment goals for MASLD [154]. A study evaluating Resmetirom, an approved pharmacotherapy for MASH, found that lower baseline Health-Related Quality of Life (HRQoL) scores, which include physical, social, emotional, and mental health components, including depression, were independently associated with female gender [293].

These findings suggest a typically female dietary pattern that may contribute to the pathogenesis of MASLD [12], especially after menopause. Therefore, it is critical to consider gender differences when optimizing recommendations and adherence strategies for postmenopausal women to ensure that interventions are tailored to address the unique challenges they face at this stage of life.

## 7. Pharmacological Treatment of MASLD

Although targeted pharmacotherapy for adults with cirrhotic MASH is not yet available, the Food and Drug Administration (FDA) recently approved resmetirom, a liver-targeted thyroid hormone receptor agonist for non-cirrhotic MASH in adults with significant liver fibrosis (stage ≥ 2). This drug has shown histologic efficacy in reducing steatohepatitis and fibrosis, with an acceptable safety profile [2]. In the phase 3 trials, MAESTRO-NAFLD-1 [294] and MAESTRO-MASH [295], resmetirom effectively reduced liver steatosis in non-cirrhotic MASH adults with significant liver fibrosis [296].

GLP1-RAs are approved for the treatment of T2D and obesity and have been explored as a potential treatment option for MASLD due to the overlap in pathophysiology [2,297,298]. Although not approved for MASLD, guidelines support their use in the treatment of T2D and obesity in MASLD to improve cardiometabolic outcomes [2]. It has been suggested that hepatic histologic benefits may be expected due to the substantial weight loss induced by this class of drugs, although this has not yet been documented [2], as well as from their pleiotropic activity.

### The Emerging Role of GLP1-RAs in MASLD

GLP1-RAs have shown promise in improving liver health through their metabolic regulatory effects, benefiting individuals with MASLD/MASH and associated CV risks [126,297,299]. They help reduce liver enzymes, hepatic fat content, inflammation, and fibrosis [300], confirming their safety and supporting their use in treating T2D and/or obesity in MASLD patients [2].

Although the exact mechanisms are not fully understood, the efficacy of GLP1-RAs in MASLD is related to the amelioration of metabolic defects associated with MetS [9,297] by improving glucose and lipid levels, hepatic insulin resistance, mitochondrial function, and regulation of insulin and glucagon levels [298,300]. Their hepatoprotective effects result from both indirect mechanisms, such as weight loss, and direct effects on the CNS, pancreas, liver, and gut through GLP-1R receptor interactions [126,300], which explain the neuroprotective, anti-inflammatory, cardioprotective, and metabolic effects of GLP1-RAs in conditions associated with MASLD [301].

GLP1-RAs activate receptors in the hypothalamus and hindbrain, including the paraventricular nucleus (PVN), dorsal medial nucleus of the hypothalamus and ARC, as well as the nucleus of the NTS, increasing satiety, reducing food intake [298,300,302], and potentially reducing anticipatory food reward [299,303]. Recent research provides evidence that GLP1-RAs appear to be effective in reducing BED in preclinical models and in adults with subclinical or clinical BED [303,304,305,306]. They also increase sympathetic nervous system activity, promoting BAT activation and white adipose tissue browning [300], contributing to significant weight loss observed in patients treated with GLP1-RAs [126,299,301]. GLP1-RAs contribute to satiety by reducing gastric emptying and motility, which helps reduce food intake [301]. However, this effect diminishes over time, while their more pronounced effect on weight and glucose control persists [299].

GLP1-RAs also exert beneficial effects on the pancreas, including increased insulin secretion, improved β-cell survival, increased glucose transporter expression, and decreased glucagon secretion [301]. These actions protect the liver from ectopic lipid accumulation by promoting nutrient storage in adipose tissue and glycogen synthesis [299,300] while reducing lipolysis. This results in less FFAs flowing to the liver, reducing substrate availability for DNL [300], HGP [297], and the secretion of TG-rich VLDL [301], while promoting fatty acid oxidation. Together, they prevent ectopic fat accumulation [307].

While it is believed that GLP1-RAs do not directly interact with hepatocytes [126,300,301], it has also been reported that their effects on lipid metabolism [299] may be influenced by GLP-1 receptors (GLP-1R) in the liver [307,308,309,310], particularly in liver endothelial cells and T cells, as shown in preclinical models [309] and liver biopsies from MASH patients [310]. Therefore, while weight loss alone has been associated with improvements in hepatic steatosis and significantly contributes to the resolution of MASH after treatment [300,301], additional, as yet unexplored mechanisms may be involved in the anti-inflammatory and anti-steatotic effects of GLP1-RAs, which require further investigation [299].

Improving systemic inflammation, a key feature of MetS [307], has been proposed as a unifying mechanism for the beneficial effects of GLP1-RAs in multiple organs [301], particularly in the liver, where they reduce the progression of MASLD and the development of HCC while providing CV protection [307]. The immunoregulatory effects of GLP1-RAs are associated with the suppression of pro-inflammatory pathways such as JNK and NF-kB, the reduction of inflammatory mediators such as CRP, TNF-alpha, IL-1β, and IL-6, and the induction of anti-inflammatory cytokines [301,307]. GLP1-RAs regulate adiponectin synthesis and decrease leptin production [301], improving insulin sensitivity and reducing inflammation, ultimately decreasing hepatic fat accumulation and the progression of fibrosis [307]. In addition, GLP1-RAs promote macrophage polarization toward an anti-inflammatory M2 phenotype [299,301,307], further helping to reduce MASH and fibrosis [297]. This may also explain their beneficial effects on the intestinal barrier and gut microbiota [301,311].

As reported in the literature [299,301,312], a growing number of randomized clinical trials (RCTs) have investigated the hepatoprotective effects of incretin-based therapies in the treatment of MASLD or MASH. These studies suggest improvements in steatohepatitis, reduced progression of fibrosis, and decreased liver-related complications [297,299], supporting these therapies for the treatment of T2D and/or obesity in patients with MASLD [313,314,315,316]. A recent meta-analysis of phase 2 RCTs confirmed these findings, showing a reduction in liver enzyme levels, decreased liver fat content, and histologic improvement in MASH without worsening fibrosis [317]. A Phase 2 study of subcutaneous semaglutide in patients with biopsy-confirmed MASH and liver fibrosis demonstrated significant MASH resolution without fibrosis progression. While the effect on fibrosis regression remains uncertain, improvements in non-invasive markers of liver fibrosis were observed [314]. These results were also confirmed in studies combining semaglutide with lipogenesis inhibitors [318,319] and in preliminary results from the ongoing phase 3 ESSENCE study [320,321].

Although not specifically indicated for MASH [2], GLP1-RAs have also reduced the risk of composite liver disease (including cirrhosis or HCC) in patients with diabetes [322,323,324], even in those with compensated cirrhosis from MASLD [325]. Preclinical data suggest that GLP1-RAs may reduce the incidence of HCC associated with MASH by restoring the balance between adiponectin and leptin levels [307,326].

Considering the high metabolic burden in MASLD, as highlighted in the ESSENCE baseline analysis [321], all studies of GLP1-RAs have confirmed their strong beneficial effects on metabolic parameters, particularly body weight, glycemic control, and plasma lipid profile. These improvements have been associated with histologic improvements in MASH [299]. Due to their pleiotropic activity, GLP1-RAs are effective not only in managing MASLD and long-term liver-related outcomes [301], but also in addressing the metabolic dysfunctions underlying MASLD, such as T2D, MetS, obesity, and CV disease [297,301,307], all of which are key to diagnosis.

Given the strong association between MetS and MASLD, as well as the increased prevalence of both conditions in postmenopausal women, GLP1-RAs are proving to be particularly effective in preventing and improving prognosis in this population, underscoring the importance of early intervention during the menopausal transition.

## 8. Do Sex Differences in GLP1-RA Response Suggest That Estrogens Enhance Efficacy in Women?

Recognizing the subgroups that would benefit from GLP-1 analog therapy would allow healthcare professionals to optimize the use of these treatments, particularly with respect to sex differences.

Studies suggest that women experience greater weight loss than men after treatment with GLP1-RAs [327,328]. Women also experienced greater improvements in glycemic control, with higher reductions in HbA1c levels with combined exenatide and metformin therapy, along with greater improvements in lipid profiles, insulin resistance, and increased adiponectin levels [329]. A retrospective analysis of the long-term effects of treatment with GLP1-RAs identified male sex as a predictor of treatment failure in terms of HbA1c. Similarly, an analysis of liraglutide found that female sex was a predictor of better glycemic response [328].

One possible explanation for the greater efficacy of GLP1-RAs in reducing body weight is the lower average body weight of women compared to men, resulting in greater drug exposure [327,328,330]. However, one study found that, in addition to female sex, a higher baseline BMI was also associated with greater weight loss after treatment with liraglutide [331]. Pharmacokinetic analysis of liraglutide exposure showed 32% higher exposure in women than in men, independent of body weight, suggesting that female sex is an independent factor promoting weight loss [332]. This raises the hypothesis that sex hormones may influence the action of GLP1-RAs and their efficacy [327], with women experiencing different effects depending on their menopausal status.

In a retrospective, real-world study evaluating the sex-specific response to liraglutide 3.0 mg in individuals with obesity without T2D, greater weight loss (in kg and %) and improvements in lipid profile and FIB-4 were observed in men compared with women [333]. These findings, observed in a sample with an average age of 50, have been linked to changes in body composition and cardiometabolic risk factors during the perimenopausal period [333,334]. Although the sample size was small, the results suggest that menopausal status may be a predictor of a reduced response to these medications.

In fact, sex differences in the response to GLP-1 agonists have been linked to levels of sex steroids [264,335]. Therefore, while women generally respond well to GLP1-RAs [327], their effectiveness seems to decrease after menopause [328,336]. Consistent with this, early initiation of liraglutide therapy was associated with better glycemic control, as was female sex [336].

The anorexigenic effect of liraglutide is primarily mediated by a central effect, with a secondary contribution from inhibition of gastric emptying. Both mechanisms can be influenced by estrogen.

GLP-1 acts on brain areas involved in eating behavior, such as the ARC and reward circuitry, which are also targets of estrogen’s anorexigenic action. This may provide a neuroanatomical basis for the interaction between the GLP-1R and sex [337].

Changes in endogenous GLP-1 levels have been observed during BED episodes [303], and the increased consumption of palatable foods after ovariectomy was reversed by estrogen replacement [259,338,339,340]. This supports the likely role of sex hormones in the regulation of eating behavior mediated by GLP1-RAs [341]. For example, it has been observed that female rats are more sensitive than males to the anorectic effects of centrally administered GLP1-RA (exendin-4), and this sex difference is abolished by anti-estrogens. The effect of GLP-1 on feeding behavior is based on central signaling through the estrogen receptor ERα, which is necessary for its effects on food reward [342]. Additionally, given the link between depression and binge eating, studies of GLP1-RA treatment for T2D have shown that women experience a greater reduction in anxiety and depression risk compared to men. This suggests an interaction between GLP-1R, the central and peripheral nervous systems, and estrogen in anorexigenic actions [343].

In addition, some researchers have hypothesized that sex hormones may influence GLP-1 efficacy, as the removal of ovarian estrogen may affect GLP-1 production [328]. For example, it has been reported that estrogen increases total GLP-1 secretion from human pancreatic alpha cells and intestinal L-cells, suggesting that sex may influence endogenous intestinal or pancreatic responses to GLP-1 [327,344]. In premenopausal women, endogenous responses to GLP-1 appear to change in relation to fluctuations in gonadal hormones [327]. In ovariectomized mice, glucose tolerance and insulin secretion were impaired, and intestinal and pancreatic GLP-1 secretion was reduced. This effect was reversed by E2 supplementation, confirming the essential role of sex hormones in maintaining glycemic homeostasis [344]. Similar to GLP1-RAs, E2 has been shown to prevent the exacerbation of diabetes by improving glucose-stimulated insulin secretion in a prediabetic mouse model [196], probably through GLP-1-mediated regulation. This may explain the increased risk of T2D after menopause, with further exacerbation associated with lower premenopausal E2 levels [85,196].

Finally, although gastric emptying rates tend to be slower in women, it has been hypothesized that ovarian hormones may influence this process by activating gastric vagal afferents through GLP1-RAs [264,327]. However, further research is needed to confirm this.

Overall, while it remains unclear whether GLP1-RAs have sex-specific effects on reducing CV risk or influencing parameters like WC, blood pressure, and the frequency of MACE [328], it is important to consider the possibility that biological sex may positively modulate treatment response [264,328].

This area of investigation is critical in the context of recommending the use of GLP1-RAs for the treatment of obesity and/or T2D to improve cardiometabolic outcomes associated with MASLD/MASH [2]. The increased incidence of T2D and obesity in postmenopausal women has led to increased use of GLP1-RAs. At the same time, given the bidirectional relationship of these conditions with MASLD, further research should investigate the efficacy of GLP1-RAs in this population.

## 9. Combining Estrogens with GLP1-RAs: A Promising Dual Therapy for the Prevention of MASLD and Metabolic Dysfunction in Postmenopausal Women?

Pharmacological intervention in chronic diseases often requires a polypharmacological approach. For example, the combination of GIP with GLP1-RAs has been shown to significantly improve MASH, reduce fibrosis, and decrease liver steatosis [345,346,347], while also having a synergistic effect on cardiometabolic risk factors [348,349,350]. This combination strategy enhances treatment efficacy for the management of MASLD/MASH as a multisystemic disease [351], by targeting multiple metabolic pathways.

Estrogen signaling plays a critical role in regulating energy balance throughout the brain and body, but these effects diminish after menopause, increasing the risk of metabolic disorders and higher mortality [265]. The absence of estrogen leads to significant changes in adipose tissue, including alterations in adipocytes, lipid metabolism, insulin sensitivity, adipokine secretion, and inflammation [352], factors involved in MASLD. However, these changes have been reversed with GLP1-RAs, such as liraglutide [352].

Given the role of estrogen in modulating GLP-1 action, the estrogen-GLP-1 pathway may be an important therapeutic target [353], especially in conditions such as obesity and MetS during estrogen deficiency [352]. Combining GLP1-RAs with estrogen during the menopausal transition could help counteract the loss of estrogen’s protective effects against the development of MASLD, the hepatic manifestation of MetS, and enhance the efficacy of GLP1-RAs, particularly if treatment is initiated early.

Estrogen replacement has been shown to be effective in preventing the development and progression of liver disease [5,12,26], and the conjugation of GLP-1 with 17-β-estradiol was one of the first peptide fusion experiments [351]. However, the oncogenic and gynecologic side effects associated with estrogens have limited the use of GLP-1 conjugates and estradiol. On the other hand, weight loss induced by GLP-1 alone is often insufficient [264], particularly when addressing the multiple metabolic changes associated with menopause due to estrogen deficiency.

To overcome these shortcomings, a stable GLP-1-estradiol conjugate was developed using unimolecular polypharmacy [264]. The use of a peptide transporter that specifically delivers estrogen to cells expressing the GLP-1R receptor allows targeted estrogen delivery to GLP-1R-expressing tissues without causing unwanted side effects such as endocrine toxicity or tumorigenesis. In addition, this selective targeting has proven to be significantly more effective than activating individual GLP-1 or estrogen receptors, thereby maximizing their efficacy [354]. In mice with MetS, estrogen was shown to greatly enhance the metabolic benefits of components of MetS (adiposity, hyperglycemia, and dyslipidemia) compared to single hormones by exploiting its pleiotropic effects on energy, glucose, and lipid metabolism. The reduction in body weight was associated with appetite suppression and reduced food intake. The loss of fat mass was also associated with decreased leptin levels, suggesting an improvement in leptin sensitivity. Positive effects on glycemic control, insulin sensitivity, dyslipidemia, and respiratory quotient were also observed [354].

The synergistic action of these two hormones in activating brain areas that regulate food intake has been shown to exceed the anorectic effects of GLP-1 alone, with greater efficacy in the New Zealand obese mouse model of T2D. The anorectic signaling of the GLP-1-E2 conjugate results in significantly higher expression of POMC neurons, leading to a significant reduction in body weight. In addition, the suppression of hyperphagia has protected beta cells from glucolipotoxic damage induced by carbohydrates, improving glucose tolerance and insulin sensitivity, thus preventing the onset of T2D [355].

The GLP-1-E2 dual agonist has also shown superior efficacy compared to the GLP-1 monomer in preventing insulin-deficient T2D (insulin sensitivity and glucose homeostasis) in wild-type mice. This was observed independently of fat mass improvement and without any gynecological effects of E2. It has also shown transcriptional activity in the mouse pancreatic beta cell line and an improvement in glucose-stimulated insulin secretion (GSIS) in static incubation in cultured human islets [356]. Another study showed that the conjugate ameliorated insulin-deficient diabetes in male mice to a greater extent than a single GLP-1 agonist, without causing feminizing effects, and upregulated anti-apoptotic pathways in cultured human islets produced by the monoagonists [357]. In fact, the antidiabetic effect of the dual agonist has been recently linked to the enhancement of the insulinotropic effects of GLP-1 [351,358]. Maximizing estrogen signaling to β-cells via GLP-1 has been shown to reduce daily insulin needs by 60%, increase β-cell survival, and protect against cytokine-induced dysfunction in mouse models of T2D and in human micro-islets [358]. Based on these findings, the authors suggest that the dual agonism of E2 and GLP-1 may be more effective than either monotherapy in improving β-cell function in both mice and humans.

These metabolic benefits may explain the positive effects on liver health observed in these preclinical studies, which reported significant reductions in lipid content in both liver and adipose tissue [355] and improvements in hepatosteatosis and hepatocellular damage [354]. Since the potential role of receptors in these tissues remains uncertain, it is likely that these results are due to the central actions of the conjugate. By inducing severe caloric restriction, weight loss, and improving insulin sensitivity, it positively influences the overall metabolism of the body [355].

Another example is the use of this dual agonist in the treatment of PCOS, the leading cause of hyperandrogenism in women during their reproductive years, where disrupted estrogen signaling and elevated androgen levels lead to impaired hepatic metabolism [177,359]. This condition is strongly associated with an increased risk of MASLD [22,102,175]. At relatively low doses, GLP-1-E2 has demonstrated beneficial effects in a PCOS mouse model of obesity, particularly in addressing metabolic dysfunction, including reductions in body weight and fat, as well as improvements in glucose management and insulin resistance, without causing uterine effects. These benefits were found to be superior to GLP-1/GIP and were primarily attributed to effects on food intake at the brain level. In addition, changes in the hypothalamic proteome related to inflammation, apoptotic processes, autophagy, and vesicular trafficking have been suggested as contributing factors [360]. Activation of autophagy may be a promising strategy for the prevention and treatment of metabolic diseases [360,361].

Although peripheral effects of the GLP-1-E2 conjugate are expected, its metabolic effects appear to be primarily due to its central action, which suppresses not only food intake but also food reward [264,268].

As noted above, given the involvement of sex hormones in food regulation, BEDs are more common in women than in men and become even more prevalent during menopause. Therefore, targeting both GLP-1 and E2 signaling may represent a novel strategy to improve the treatment of obesity and food reward regulation [264,303] in these women.

E2 has been reported to enhance central GLP-1-induced suppression of food intake in several brain regions, including the hypothalamic PVN and other areas such as the supramammillary nucleus (SUM) and medial amygdala (MeA) [268].

In addition to these regions, the GLP-1-E2 conjugate transports bioactive estrogens to the dorsal raphe nucleus (DRN), a key area involved in the development of eating behavior [362]. Research has shown that the replacement of 17β-estradiol in ovariectomized female mice suppresses food intake by interacting with ERα in serotonin (5-HT) neurons in the DRN [363]. This suggests that some of the effects of the co-agonist on BED are mediated by estrogens [99]. Research using functional neuroimaging techniques has confirmed that the anti-obesity activity of the GLP-1 E2 dual agonist occurs through interactions with co-expressed receptors in brain areas that regulate homeostatic eating behavior and reward. Specifically, in mouse models, the SUM was identified as a direct target site for the synergistic activity on reward, contributing to a greater reduction in body weight. In addition, more traditional areas involved in energy balance regulation, such as the lateral hypothalamus (LH) and, to a lesser extent, the NTS, mediate the metabolic benefits of the co-agonist on food intake and body weight regulation without affecting food-motivated eating behavior [335]. This study also confirmed that the GLP-1-E2 combination was more effective than the individual hormones used separately [335].

The MeA expresses significant GLP-1Rs, in addition to ERα, and the GLP-1-E2 conjugate has been shown to release bioactive estrogens in this region in mice. This suggests that the expression of ERα in MeA neurons is at least partially responsible for mediating the weight-loss effects of the dual agonist. ERα signaling in these neurons has been linked to the stimulation of physical activity, which promotes energy expenditure [362]. As a result, this population of extra-hypothalamic receptors may interact with other neural networks and contribute to a broader range of actions related to body weight reduction. In addition, the presence of estrogens appears to be essential for mediating the effect of GLP-1 on food intake in the PVN nucleus [364].

In general, this stable conjugate enhances the benefits of energy balance, insulin secretion, and β-cell protection, while binding estrogen to prevent its release into the bloodstream. This mechanism helps to specifically target cells that express the GLP-1R, thus avoiding unwanted hormonal effects [264], such as the uterotrophic and oncogenic effects seen in breast cancer models [264,354,365]. In contrast, the labile conjugate releases extracellular estrogen [264,354], which may lead to potential side effects. Moreover, the stable conjugate does not appear to interfere with the hypothalamic–pituitary–gonadal axis [264].

This approach can address all metabolic, physiological, and psychological changes associated with menopause and offers a potential new strategy for the prevention/treatment of MASLD/MASH and its comorbidities (T2D, CV disease, MetS, mental disorders) (Figure 2).

## 10. Conclusions

MASLD is a complex disease influenced by multiple genetic, lifestyle, and environmental factors. Previous studies have shown a sex difference in the prevalence of MASLD, generally higher in men but reversing after menopause, highlighting the critical role of hormonal interactions in liver disease risk. The loss of estrogen protection during menopause calls for more personalized therapies.

Menopause negatively affects MASLD and its progression due to the complex interaction between estrogen deficiency, visceral fat, and glucose and lipid metabolism dysfunction, which exacerbates the increased risk of CV disease observed after menopause. In addition, estrogen deficiency is associated with physiological and psychological changes that contribute to uncontrolled eating and depressive disorders, further promoting a vicious cycle of increased adiposity and metabolic dysfunction. Weight loss in this group can improve quality of life, self-esteem, and psychological well-being, while also helping to prevent metabolic diseases such as MASLD.

Given the recommendation of GLP1-RAs for the treatment of obesity and T2D in MASLD patients and the positive modulation of these agents by estrogens, which decline after menopause, a combined therapeutic approach may provide a more targeted therapeutic strategy in this subgroup. Although preclinical data are limited, a stable GLP-1–estrogen conjugation has shown promising synergistic effects, particularly in reducing food intake, food reward, and episodes of uncontrolled eating. By acting primarily on the CNS, this synergy promotes weight loss and helps address excess adiposity, a major contributor to MAFLD, MetS, and related diseases. In addition, this combination may improve insulin sensitivity, β-cell function, glucose and lipid metabolism, and leptin resistance, potentially preventing or reversing metabolic dysfunction and T2D, while providing hepatoprotective benefits. This suggests that this pharmacotherapy may be an effective strategy for the prevention of MASLD and its progression after menopause.

## 11. Future Directions

The physiological and psychological changes that occur after menopause significantly influence the epidemiology, pathogenesis, and associated complications of MASLD. This highlights the need to stratify patients based on these factors to guide more targeted therapeutic approaches, making this a critical area for future research.

The combination of GLP-1 and estrogen in a stable conjugate may offer greater efficacy than GLP1-RAs alone in treating metabolic dysfunction in postmenopausal women, improving overall metabolic health without unwanted hormonal interference. However, despite the promise of this dual agonist approach, studies to date have been limited to preclinical models such as mice and cultured micro-islets, leaving a significant gap in the clinical literature. Given the current lack of clinical evidence on the efficacy and safety of this dual agonist approach in this subpopulation, there is a critical need to explore this mechanism in a large-scale trial. This review aims to highlight the importance of further investigation and validation of these findings in the clinical setting to address this research gap.

## Figures and Tables

**Figure 1 biomedicines-13-00855-f001:**
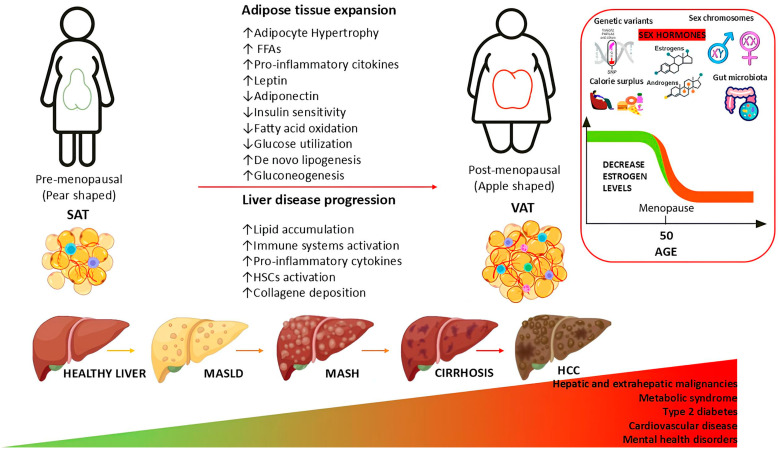
MASLD development and progression after menopause: the interplay between hormonal changes, weight gain, metabolic dysfunction, genetics, and mental health disorders. SAT: subcutaneous adipose tissue; FFAs: free fatty acids; VAT: visceral adipose tissue; HSCs: hepatic stellate cells; MASLD: metabolic dysfunction-associated steatotic liver disease; MASH: metabolic dysfunction-associated steatohepatitis; HCC: hepatocellular carcinoma.

**Figure 2 biomedicines-13-00855-f002:**
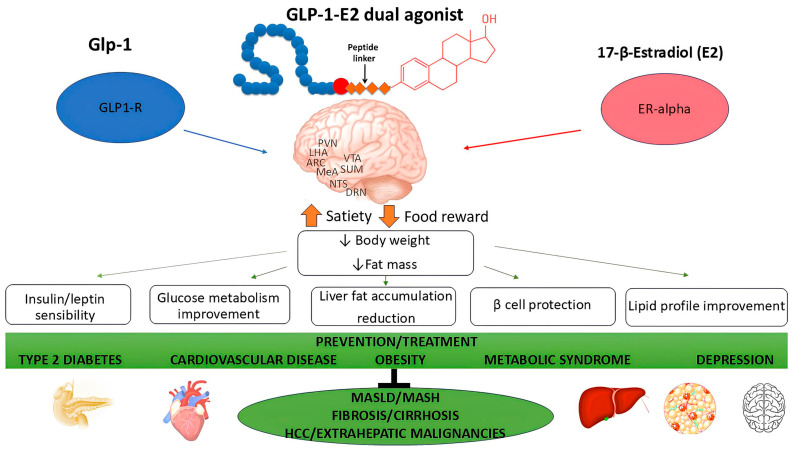
Stable GLP-1–estrogen conjugate as a promising therapeutic strategy for MASLD and its hepatic and extrahepatic comorbidities (T2D, CV diseases, MetS, mental disorders). Glp-1: glucagon-like peptide 1; GLP1-R: glucagon-like peptide 1 receptor; ER-alpha: estrogen receptor alpha; PVN: paraventricular nucleus; LHA: lateral hypothalamic area; ARC: arcuate nucleus; MeA: medial amygdala; VTA: ventral tegmental area; NTS: nucleus of the solitary tract; DRN: dorsal raphe nucleus; SUM: supramammillary nucleus; MASLD: metabolic dysfunction-associated steatotic liver disease; MASH: metabolic dysfunction-associated steatohepatitis; HCC: hepatocellular carcinoma.

## Data Availability

The original contributions presented in this study are included in the article. Further inquiries can be directed to the corresponding author.

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
