# Peer review of "MASLD: Prevalence, Mechanisms, and Sex-Based Therapies in Postmenopausal Women"

_biomedicines, 2025, doi:10.3390/biomedicines13040855_

Round 1
Reviewer 1 Report
Comments and Suggestions for Authors
This manuscript highlights the prevalence of MASLD, its comorbidities (type 2 diabetes T2D, CV, mental disorders), pathogenetic mechanisms, and pharmacological treatment with 24 GLP-1 receptor agonists (GLP1-RAs), focusing on postmenopausal women. This is a critical topic. The manuscript is well-structured and effectively presents key points.
I have two important points:
1. Authors refer to "MASLD" as Metabolic steatotic liver disease. The more widely accepted term is Metabolic dysfunction-associated steatotic liver disease (MASLD).
2. Authors don’t clearly state the primary objective. Consider explicitly mentioning that the review analyzes the potential of GLP-1-estrogen conjugates as a therapeutic approach.
Author Response
Reviewer 1
Comments and Suggestions for Authors
This manuscript highlights the prevalence of MASLD, its comorbidities (type 2 diabetes T2D, CV, mental disorders), pathogenetic mechanisms, and pharmacological treatment with 24 GLP-1 receptor agonists (GLP1-RAs), focusing on postmenopausal women. This is a critical topic. The manuscript is well-structured and effectively presents key points.
I have two important points:
- Authors refer to "MASLD" as Metabolic steatotic liver disease. The more widely accepted term is Metabolic dysfunction-associated steatotic liver disease (MASLD).
- Authors don’t clearly state the primary objective. Consider explicitly mentioning that the review analyzes the potential of GLP-1-estrogen conjugates as a therapeutic approach.
Author's reply to reviewer's report:
We greatly appreciate the reviewer's insightful and constructive comments, which helped us improve the paper and its primary objective.
We address all of the reviewer's concerns here, providing a point-by-point response to the reviewer's comments and highlighting them in yellow in the revised version of the manuscript.
Comment 1. Authors refer to "MASLD" as Metabolic steatotic liver disease. The more widely accepted term is Metabolic dysfunction-associated steatotic liver disease (MASLD).
Response 1: We thank the reviewer for bringing this comment to our attention. We apologize and have updated the correctly accepted term "Metabolic dysfunction-associated steatotic liver disease" in the abstract (lines 13 and 45) as suggested.
Comment 2. Authors don’t clearly state the primary objective. Consider explicitly mentioning that the review analyzes the potential of GLP-1-estrogen conjugates as a therapeutic approach.
Response 2: We thank the reviewer for this suggestion and fully agree with this observation. Therefore, the following corrections have been made to the Abstract (lines 26-36) and the Introduction (lines 84-92) to better reflect the primary objective of the review.

Reviewer 2 Report
Comments and Suggestions for Authors
The authors aimed to highlight the importance of considering factors such as sex, hormonal status, and physiological and psychological changes associated with menopause in the management of MASLD, while exploring its prevalence, pathogenetic mechanisms, and the efficacy of GLP1-RAs pharmacotherapy. This review article is extensively written and presents novel information regarding MASLD. The manuscript is well organized, scientifically sound, and comprehensively described.
Some minor remarks are as following:
- The sentence should not start with the abbreviation (e.g., MASLD).
- The abbreviations when first mentioned in the text should be written in full (e.g., T2D, GLP1-RAs, EASL-EASD-EASO, MetS).
- The abbreviations when first introduced should be used consistently thereafter throughout the text (e.g., VAT, BMI).
Author Response
Reviewer 2
Comments and Suggestions for Authors
The authors aimed to highlight the importance of considering factors such as sex, hormonal status, and physiological and psychological changes associated with menopause in the management of MASLD, while exploring its prevalence, pathogenetic mechanisms, and the efficacy of GLP1-RAs pharmacotherapy. This review article is extensively written and presents novel information regarding MASLD. The manuscript is well organized, scientifically sound, and comprehensively described.
Some minor remarks are as following:
- The sentence should not start with the abbreviation (e.g., MASLD).
- The abbreviations when first mentioned in the text should be written in full (e.g., T2D, GLP1-RAs, EASL-EASD-EASO, MetS).
- The abbreviations when first introduced should be used consistently thereafter throughout the text (e.g., VAT, BMI).
Author's reply to reviewer's report:
We sincerely thank the reviewer for taking time to review this paper and for providing valuable comments and helpful feedback. We have addressed all concerns in the revised manuscript, and below we provide a point-by-point response to the reviewer's comments, which are highlighted in yellow in the revised manuscript.
Comment 1: The sentence should not start with the abbreviation (e.g., MASLD).
Response 1: We strongly agree with the reviewer's suggestion and have changed the abbreviation to the full term "Metabolic dysfunction-associated steatotic liver disease" followed by the abbreviation (MASLD) (line 45).
Comment 2: The abbreviations when first mentioned in the text should be written in full (e.g., T2D, GLP1-RAs, EASL-EASD-EASO, MetS).
Response 2: We thank the reviewer for this important observation. We have changed the abbreviations to the full term: Type 2 diabetes (T2D) line 54, metabolic syndrome (MetS) line 59, GLP-1 receptor agonists (GLP1-RAs) line 87, European Association for the Study of the Liver (EASL)-European Association for the Study of Diabetes (EASD)-European Association for the Study of Obesity (EASO) (EASL-EASD-EASO) line 104-105, cardiovascular (CV) line 124.
Comment 3: The abbreviations when first introduced should be used consistently thereafter throughout the text (e.g., VAT, BMI).
Response 3: We thank the reviewer for pointing this out and we checked the text and we have used the abbreviations throughout the full text as suggested for body mass index BMI, visceral adipose tissue VAT, triglycerides TG, high-fat diet HFD.

Reviewer 3 Report
Comments and Suggestions for Authors
The manuscript provides a thorough and well-researched review of MASLD, highlighting key aspects of sex differences in disease progression, pathogenesis, and treatment options. The discussion is supported by an extensive range of references, and the inclusion of mechanistic insights and therapeutic implications adds significant value to the field. The writing is overall clear and well-structured, and the manuscript successfully integrates various factors influencing MASLD.

Author Response
Reviewer 3
Comments and Suggestions for Authors
General Assessment:
The manuscript provides a thorough and well-researched review of MASLD, highlighting key aspects of sex differences in disease progression, pathogenesis, and treatment options. The discussion is supported by an extensive range of references, and the inclusion of mechanistic insights and therapeutic implications adds significant value to the field.
The writing is overall clear and well-structured, and the manuscript successfully integrates various factors influencing MASLD. However, there are a few aspects that could be improved to enhance readability and clarity. Below, I outline specific suggestions that I believe will strengthen your manuscript.
Author's reply to reviewer's report:
We sincerely thank the reviewer for his valuable time and effort in reviewing our manuscript. We have carefully read the comments and have revised the manuscript according to the reviewer's thoughtful and constructive suggestions, which have helped to improve the quality and novelty of our manuscript. We hope that we have addressed all of the reviewer's concerns. We provide a detailed point-by-point response to the reviewer's suggestions, and all changes are highlighted in yellow in the revised paper.
Comments 1:
- Section numbering format: The current numbering system (e.g., 1.0, 1.1, etc.) is confusing and difficult to follow. I recommend revising the section titles to a standard format using whole numbers (e.g., 1. Introduction, 2. Sex Differences in MASLD, 3. Gender Differences in Extrahepatic Diseases, etc.) for improved readability and clarity.
- Response 1: We strongly appreciate the reviewer's valuable suggestion. We have revised the section titles as recommended using only whole numbers (e.g. 1. Introduction) for section headings (line 44, line 94, line 179, line 416, line 764, line 808, line 924, line 1026, line 1107, line 1245, line 1272), and rewriting subparagraphs in italics without numbers (line 194, line 304, line 381, line 449, line 498, line 603, line 708, line 939). We hope that these changes will improve the clarity and readability of the text.
Comments 2:
- Abbreviation Usage: Abbreviations introduced in the abstract should be redefined the first time they appear in the main text to ensure clarity for the reader.
- Response 2: We strongly agree with the reviewer's comment and have also included the full term followed by the abbreviation in the main text, e.g. for type 2 diabetes (T2D) line 54, metabolic syndrome (MetS) line 59, GLP-1 receptor agonists (GLP1-RAs) line 87, cardiovascular (CV) line 124.
Comments 3:
- Abbreviation Usage: While key abbreviations are appropriately defined, their excessive use may hinder readability. Some abbreviations, such as NZO, ALD and HR, appear only once in the text and do not contribute significantly to readability; consider removing these to reduce unnecessary complexity.
- Response 3: We thank the reviewer for this suggestion, and we have deleted the abbreviations that are mentioned only once in the text as suggested, such as alcohol-related liver disease (ALD) line 133, Model for End-Stage Liver Disease (MELD) line 162, hazard ratio (HR) lines 226-227, ST-elevation myocardial infarction (STEMI) line 328, non-ST-elevation myocardial infarction (N-STEMI) line 329, ischemic heart disease (IHD) line 338, heart failure with preserved ejection fraction (HFpEF) line 370, extracellular matrix (ECM) line 539, Sirtuin1 (SIRT1) line 548, trimethylamine N-oxide (TMAO) line 718, New Zealand obese (NZO) line 1150.
Comments 4:
- Improving Fluency and Reducing Redundancies: Some ideas, such as the relationship between MASLD, menopause, and visceral adiposity, are repeated across multiple sections. Consolidating these concepts into a single wellstructured section would help avoid redundancy and improve the manuscript’s fluency.
- Response 4: We agree with the reviewer's valuable suggestion. As a result, we have decided to combine the previous two subsections "1.3.3 Adipose tissue dysfunction after menopause" (line 605) and "1.3.4 Involvement of estrogens in adipose tissue endocrine function" (line 705) into a single paragraph entitled "Involvement of estrogens in adipose tissue dysfunction after menopause" (line 603). The changes we have made are from line 603 to line 707. We hope that this revision will improve the flow of the text and enhance readability, as suggested.
Comments 5:
- Stronger Justification for GLP-1 + Estrogen Combination Therapy: While the manuscript mentions preclinical studies on GLP-1 receptor agonists combined with estrogen, it remains unclear whether any ongoing clinical trials support this therapeutic approach. Providing references to ongoing studies or acknowledging the current research gap would strengthen this discussion.
- Response 5: We strongly agree with the reviewer's important comment and apologize for the potential confusion. We have clarified that studies of this dual conjugate have been performed in preclinical models as indicated in lines 1158-1160, 1163-1164, 1168-1170, 1171-1172 and lines 1182, 1209 and 1217 to avoid any confusion. In addition, we have separated the "Conclusion" and "Future Directions" into two separate paragraphs. The "Conclusion" section has been revised from lines 1259 to 1270. In the "Future directions" section (lines 1272-1286), we have emphasized the importance of further investigations to potentially translate and validate these findings in the clinical setting, thereby addressing this research gap.
Comments 6:
- Diagnostic Tools: The manuscript could expand on non-invasive diagnostic tools, such as elastography, emphasizing their applicability in postmenopausal women.
- Response 6: We thank the reviewer for this valuable suggestion. We have added a paragraph entitled "Applicability of non-invasive diagnostic tools for screening for postmenopausal liver disease" (lines 764-806) to discuss the potential of non-invasive diagnostic tools in postmenopausal women and to emphasize the need for further research to assess their efficacy in this subgroup of patients.
